# Genome-Wide Identification, Characterization and Expression Analysis of the JAZ Gene Family in Resistance to Gray Leaf Spots in Tomato

**DOI:** 10.3390/ijms22189974

**Published:** 2021-09-15

**Authors:** Yaoguang Sun, Chunxin Liu, Zengbing Liu, Tingting Zhao, Jingbin Jiang, Jingfu Li, Xiangyang Xu, Huanhuan Yang

**Affiliations:** School of Horticulture and Landscape Architecture, Northeast Agricultural University, Harbin 150030, China; sunygneau@gmail.com (Y.S.); liucxneau@gmail.com (C.L.); liuzbneau@gmail.com (Z.L.); zhaottneau@gmail.com (T.Z.); jiangjbneau@gmail.com (J.J.); lijfneau@gmail.com (J.L.)

**Keywords:** *Solanum lycopersicum*, *JAZ* gene, gray leaf spots, functional verification

## Abstract

The plant disease resistance system involves a very complex regulatory network in which jasmonates play a key role in response to external biotic or abiotic stresses. As inhibitors of the jasmonic acid (JA) signaling pathway, JASMONATE ZIM domain (JAZ) proteins have been identified in many plant species, and their functions are gradually being clarified. In this study, 26 *JAZ* genes were identified in tomato. The physical and chemical properties, predicted subcellular localization, gene structure, cis-acting elements, and interspecies collinearity of 26 *SlJAZ* genes were subsequently analyzed. RNA-seq data combined with qRT-PCR analysis data showed that the expression of most *SlJAZ* genes were induced in response to *Stemphylium lycopersici*, methyl jasmonate (MeJA) and salicylic acid (SA). Tobacco rattle virus RNA2-based VIGS vector (TRV2)-*SlJAZ25* plants were more resistant to tomato gray leaf spots than TRV2-00 plants. Therefore, we speculated that *SlJAZ25* played a negative regulatory role in tomato resistance to gray leaf spots. Based on combining the results of previous studies and those of our experiments, we speculated that *SlJAZ25* might be closely related to JA and SA hormone regulation. *SlJAZ25* interacted with *SlJAR1*, *SlCOI1*, *SlMYC2*, and other resistance-related genes to form a regulatory network, and these genes played an important role in the regulation of tomato gray leaf spots. The subcellular localization results showed that the *SlJAZ25* gene was located in the nucleus. Overall, this study is the first to identify and analyze *JAZ* family genes in tomato via bioinformatics approaches, clarifying the regulatory role of *SlJAZ25* genes in tomato resistance to gray leaf spots and providing new ideas for improving plant disease resistance.

## 1. Introduction

Tomato (*Solanum lycopersicum*) is one of the most widely planted horticultural crop species worldwide and has obvious economic benefits. However, during the process of plant growth, various stresses (e.g., diseases, insect pests, drought, and cold) lead to a straight decline in yield and quality. Under natural conditions, plants are continually attacked by pathogens; as such, throughout the long-term struggle against pathogens, plants have evolved complex defense patterns [1]. There are many ways for plants to protect themselves, such as programmed cell death, secretion of antibacterial substances, and production of endogenous hormones (salicylic acid, jasmonic acid, ethylene) [2]. Simultaneously, for plants to better adapt to changes in their environment, evolution of their genome is particularly important [3]. Members of all kinds of gene families perform specific functions and connect to form a network that controls plant disease resistance; in addition, other factors, such as transcription and epigenetic regulation, also play an important role [4,5,6]. Discovering and further elucidating the roles of these genes is the focus of current research.

Jasmonate ZIM (Zinc-finger protein expressed in Inflorescence Meristem) domain proteins (JAZ) are important negative regulators within the jasmonic acid (JA) signaling pathway and play an important role in the plant defense response [7]. When plants are subjected to biotic or abiotic stress, JA, a natural signaling molecule, can induce endogenous JA to accumulate and activate the expression of downstream response genes [8]. Furthermore, JA participates in plant growth and development, and in defense responses, such as pathogen infection and insect injury [9,10,11]. The JA content is dynamically balanced in plants [12]. When the JA content in plants is low, JAZ proteins bind to *MYC2* transcription factors (TFs) to inhibit the transcription of downstream response genes [13]. In contrast, JAZ proteins bind JA (-Ile) receptor complexes for subsequent degradation by the 26S proteasome, enabling the transcriptional activity of *MYC2* TFs and ultimately activating the JA signaling pathway [14]. Twelve members of the *JAZ* gene family have been identified in *Arabidopsis thaliana* [15]. In addition, they have been reported in rice, maize, soybean and tobacco crop species [16,17,18,19]. In tomato, some attention has been devoted to the identification, characterization, and functional analysis of *JAZ* gene family members [20,21,22]. The members of the *JAZ* gene family are also gradually expanding, and their functions are becoming increasingly clear. Although there are some studies [23,24,25] on resistance to gray leaf spots, the role of the *JAZ* gene family in tomato resistance to gray leaf spots has rarely been studied [26].

JAZ proteins belong to a subfamily of the TIFY (containing a highly conserved TIFY motif) superfamily and are unique zinc finger proteins [27]. Current research shows that JAZ proteins contain two conserved domains: a TIFY domain located at the N-terminus and a Jas domain (CCT_2) at the C-terminus [28]. The TIFY motif of the conserved domain is TIF[F/Y]XG [29]. The Jas domain is composed of 12–29 amino acids (aa), and the conserved sequence is SLX_2_FX_2_KRX_2_RX_5_PY [30]. Previous studies have demonstrated that the TIFY domain is related to the synthesis of JAZ dimers and the binding of TF suppressors such as *NINJA* and *TOPLESS* [31,32,33]. Members of the plant *JAZ* gene family are expressed largely in response to plant stress and are key regulatory factors in the plant response to pathogenic stress [34]. The binding of JAZ proteins to COI1 depends on the Jas domain, while the Jas domain controls the degradation of the JAZ protein [35]. Compared with their wild-type counterparts, *A. thaliana JAZ5*/*JAZ10* mutants are more susceptible to *Pseudomonas syringae tomato* DC3000, which may be due to the accumulation of coronatine [36]. Similarly, mutations in *AtJAZ7* led to increased sensitivity of plants to *Fusarium oxysporum* and *P. syringae* [37]. In a study of tobacco resistance to black shank, the expression of *NtJAZ1* was higher in susceptible than in resistant plants, which indicated that *NtJAZ1* is a negative resistance factor [38]. Overexpression of *OsJAZ8ΔC* revealed that the protein encoded by this gene played a negative regulatory role in JA-mediated resistance to powdery mildew in rice [39]. Taken together, these results suggest that the expression of JAZ protein in plants may reduce plant defense against diseases, but the relationship and mechanism between them are not fully understood.

Tomato gray leaf spots are one of the most severe plant diseases worldwide. The pathogens responsible for the disease are members of *Stemphylium*, including *S. lycopersici*, *S. solani* and *S. floridanum*, which can cause damage to a variety of vegetable species [40]. Based on morphological analysis, the dominant species causing the disease was determined to be *S. lycopersici* [41]. This pathogen infects plant leaves mainly through stomata and vesicles in the substomatal cavity. In warm and humid environments (temperature of 28–32 °C and humidity greater than 80%), the disease is even worse in plants [42]. Resistance against gray leaf spots has been identified in the wild species *Solanum pimpinellifolium* and is provided by a single incompletely dominant gene, *Sm*, which is located on chromosome 11 between markers InDel-343 and InDel-FT-32 at a distance of 185 kb from each marker [43]. Unfortunately, the *Sm* gene has not been cloned thus far. Previous studies have shown significant positive correlations among the cell wall, enzyme activity, endogenous hormones, and disease resistance in the process of pathogen invasion [44]. However, the pathogenesis of gray leaf spot pathogens is not completely understood, and synergistic gene mining is particularly important.

In this study, 26 members of the *JAZ* gene family were identified in tomato for the first time, the number of which is approximately twofold greater than that identified in *A. thaliana*. The characteristics of the validated *SlJAZ* genes were investigated through phylogenetic, chromosomal location, collinearity, genetic structure, conserved protein domain and cis-acting element analyses. The spatiotemporal expression characteristics of *SlJAZ* genes in tomato vary widely. In addition, RNA sequencing (RNA-seq) analysis indicated that the expression patterns of *SlJAZ* genes in tomato differed before and after inoculation with *S. lycopersici*, and quantitative real-time PCR (qRT-PCR) data verified the RNA-seq results. The expression of almost all *SlJAZ* genes was induced by MeJA and salicylic acid (SA), which significantly increased the expression of these genes. We subsequently reduced the expression of *SlJAZ25* in plants by virus-induced gene silencing (VIGS), and the phenotypic results showed that the control (CK) plants were more susceptible than the *SlJAZ25*-silenced tomato plants. In summary, we speculate that the *SlJAZ25* gene may play a negative regulatory role in tomato resistance to gray leaf spots. Finally, by combining the results of previous research and our experimental conclusions, we clarified the regulatory relationship between *SlJAZ* genes.

## 2. Results

### 2.1. Identification and Characterization of JAZ Genes in Tomato

The tomato protein database was screened with HMMER (profile hidden Markov models) 3.0 (http://www.hmmer.org/, accessed on 21 February 2021), and the domains were verified with the SMART (simple modular architecture research tool) (http://smart.embl-heidelberg.de/, accessed 18 March 2021) and CDD (conserved domain database) (https://www.ncbi.nlm.nih.gov/cdd/, accessed 20 March 2021) online tools. Twenty-six *SlJAZ* genes were identified and found to be distributed unevenly across the 12 chromosomes in tomato, among which the *SlJAZ* gene is not present on chromosomes 2 or 5. *SlJAZ* genes are densely distributed on chromosome 1. There is only one *SlJAZ* gene on chromosomes 7, 9, 10 and 11, and there are two *SlJAZ* genes on chromosomes 3, 4 and 12. These genes were named *SlJAZ1* to *SlJAZ26*, and their locations on the chromosomes are shown in Figure 1.

The CDSs of the *SlJAZ* genes ranged in length from 186 bp (*SlJAZ6*) to 1284 bp (*SlJAZ14*), with a mean length of 737 bp. Among the *SlJAZ* proteins, *SlJAZ*6 was the smallest (61 aa), and *SlJAZ14* (427 aa) was the largest. The physicochemical properties of these *SlJAZ* genes were predicted with ExPASy (expert protein analysis system) (https://web.expasy.org/protparam/, accessed on 25 March 2021). The molecular weight (MW) of the proteins ranged from 6.96 kDa (*SlJAZ6*) to 44.86 kDa (*SlJAZ14*), and the theoretical pI values ranged from 4.99 (*SlJAZ2*) to 9.94 (*SlJAZ**20*). The instability index values were between 20.36 (*SlJAZ5*) and 93.54 (*SlJAZ**19*), and the aliphatic index values ranged from 56.51 (*SlJAZ25*) to 106.05 (*SlJAZ5*). The grand mean of hydropathicity results showed that the 26 *SlJAZ* genes encoded 25 hydrophilic proteins, and only *SlJAZ6* was hydrophobic (negative values, hydrophilic radicals; positive values, hydrophobic groups). Subcellular localization predicted that most of the members were located in the nucleus, while *SlJAZ2*, *SlJAZ10* and *SlJAZ17* were located in the cytoplasm, and *SlJAZ6* and *SlJAZ**20* were located in the chloroplast. More detailed information is provided in Appendix A.

### 2.2. Phylogenetic Relationships and Gene Structure Analysis of SlJAZs

To clarify the phylogenetic relationship of SlJAZs in tomato, twenty-Six SlJAZ proteins and twelve AtJAZ proteins were used to construct a phylogenetic tree through the neighbor joining (NJ) method in MEGA (molecular evolutionary genetics analysis) X (https://www.megasoftware.net/, accessed 5 April 2021) (Figure 2). Twenty-four SlJAZ genes were divided into five clades, and each branch contained tomato and *A. thaliana JAZ* genes.

The phylogenetic tree of the 26 SlJAZ genes was divided into five branches (Figure 3A). Genetic structure analysis revealed that seven (26.93%) of the SlJAZ genes were composed of five exons and four introns (Figure 3B). Genes with fewer than four exons were also present among these SlJAZs, of which six (23.08%) contained at least three exons, and SlJAZ2 contained only two exons. Twelve (46.15%) SlJAZ genes contained more than five exons, of which SlJAZ3 had the most; up to 15. To explore the function of SlJAZs, the conserved motifs of 26 proteins were analyzed. The results showed that all members contained TIFY and Jas motifs. In addition to two conserved blocks, we detected 11 well-conserved motifs (motifs 3–13) encoded within the SlJAZ genes (Figure 3C). In the phylogenetic tree, an additional 10 motifs were specific to branches. For example, motifs 6, 7 and 12 were identified in clade I; motifs 3 and 5 were present only in clade IV; and motifs 8 and 13 were discovered in clade V. The logos of these motifs are detailed in Appendix A. These results showed that the genes with similar positions in the phylogenetic tree also had a similar gene structure. The functional differences in the SlJAZ genes of tomato were probably due to the clade-specific distribution of conserved motifs.

### 2.3. Detection of Cis-Acting Elements in the Promoter Regions of SlJAZ Genes

Cis-acting elements were detected in the promoter regions of tomato *JAZ* genes (2000 bp upstream of the initiation codon), and a total of 39 types of cis-acting elements were identified. These cis-acting elements can be divided into four categories according to their characteristics: light-responsive, hormone-responsive, environmental-stress-related, and development-related elements. Figure 4 and Appendix A show the details of these cis-acting elements. All *SlJAZ* genes contained Box 4 elements. Except for *SlJAZ5* and *SlJAZ10*, all *SlJAZ* genes contained G-boxes, with some numbers as high as twelve. Twenty-two *SlJAZ* genes contained abscisic acid (ABA) response elements (ABREs); 15 *SlJAZ* genes contained JA response elements (CGTCA motifs); and 12 *SlJAZ* genes contained SA response elements (TCA elements; Figure 4C). These results suggest that members of the tomato *JAZ* gene family are closely related to hormone signal transduction pathways. Different types and numbers of cis-acting elements are distributed in different positions in gene promoter regions, which may be closely related to the specific functions of these genes.

### 2.4. Gene Duplication and Synteny Analysis of SlJAZ Genes

Gene duplication is an important factor in gene functional differentiation and gene amplification [4]. In this study, we detected five pairs of segmental duplication and tandem repeat events in *SlJAZ**s*: *SlJAZ8* and *SlJAZ12*, *SlJAZ10* and *SlJAZ15*, *SlJAZ11* and *SlJAZ26*, *SlJAZ16* and *SlJAZ22*, and *SlJAZ17* and *SlJAZ25* (Figure 5). These results indicate that gene duplication events play an important role in the differentiation of *SlJAZ* genes in the tomato genome.

To further identify the homologs of the 26 *SlJAZ* genes between tomato and other plant species, the synteny of *A. thaliana* and *Solanum tuberosum* (potato) plants with tomato was analyzed by MCScanX. The results showed that 9 *SlJAZ* genes were collinear with *A. thaliana* genes and that 16 *SlJAZ* genes were collinear with potato genes, as shown in Figure 6 and Appendix A. Some *SlJAZ* genes, especially *SlJAZ10*, *SlJAZ11*, *SlJAZ15*, *SlJAZ25* and *SlJAZ26*, were related to at least two pairs of homologs. We speculate that these genes might play an important role in the evolution of the *JAZ* gene family. Synteny analysis of *JAZ* genes in the three species showed strong collinearity despite the occurrence of chromosomal rearrangements or gene duplication.

### 2.5. Spatiotemporal Expression Characteristics of SlJAZ Genes in Tomato

To explore the expression of *SlJAZ* genes in tomato buds, flowers, leaves, roots, and fruits, FPKM (fragments per kilobase of exon per million fragments mapped) data from the tomato variety Heinz 1706 in TFGD (tomato functional genomic database) transcriptomic data (accession number: D004) were used. The original data are shown in Appendix A. Twenty *SlJAZ* genes were identified, and their expression profiles were analyzed, the results of which are shown in Figure 7A. Most of the *SlJAZ* genes were highly expressed in buds and fruits, including 15 genes in buds and 14 genes in fruits. Except for *SlJAZ14* and *SlJAZ24*, the other *SlJAZ* genes were expressed at low levels in leaves. Some genes (*SlJAZ8*, *SlJAZ16*, *SlJAZ22*, *SlJAZ23*, *SlJAZ24* and *SlJAZ25*) were widely expressed in three or four tissues, but others (*SlJAZ2* and *SlJAZ3*) were not expressed in those tissues. Taken together, these results indicate that the spatiotemporal expression profiles of *JAZ* gene family members vary widely in tomato.

### 2.6. Expression Patterns of SlJAZ Genes According to RNA-seq after Inoculation with S. lycopersici

To explore the response of *SlJAZ* genes to inoculation with *S. lycopersici*, we investigated 17 *SlJAZ* genes. The expression profiles of these genes among the different materials are shown in Figure 7B, and the FPKM values of these genes are listed in Appendix A. The expression levels of these genes in CK1 and CK2 were approximately the same, indicating that the expression of *JAZ* genes in resistant and susceptible varieties was similar when the pathogen was not present. After three days of inoculation with *S. lycopersici*, the expression of six *SlJAZ* genes decreased to different degrees, but the decrease was only half of that in the original. The expression of *SlJAZ7* was so low that it was almost negligible; thus, we speculate that this gene may not be involved in pathogen resistance. Interestingly, the expression of most *JAZ* genes increased significantly after inoculation with *S. lycopersici*. Compared with those in the CK group, *SlJAZ19* expression in the treatment group increased nearly 200-fold, *SlJAZ11* and *SlJAZ17* expression increased approximately 100-fold, and *SlJAZ21* and *SlJAZ25* expression increased approximately 10-fold.

### 2.7. Expression Profiles of SlJAZ Genes as Assessed via qRT-PCR

To further understand the expression patterns of the *SlJAZ* genes in tomato, qRT-PCR was used to detect the expression of *SlJAZ* genes after inoculation with *S. lycopersici* (Appendix A) or exogenous application of JA (Appendix A) and SA (Appendix A). Based on the RNA-seq data and phylogenetic trees, we selected 16 differentially expressed *SlJAZ* genes. The qRT-PCR results validated the reliability of the RNA-seq data. After inoculation with *S. lycopersici*, the expression patterns of 10 genes tended to increase, and those of 6 genes tended to decrease (Appendix A). Twenty-four hours after inoculation with *S. lycopersici*, the expression of *SlJAZ20* was 77.09 times higher than that in the CK, and twelve hours after inoculation with *S. lycopersici*, the expression of *SlJAZ25* increased 4.81-fold (Figure 8). The fluctuation in the expression of these genes indicated that the selected *SlJAZ* genes were closely involved in *S. lycopersici* infection. Notably, when we treated plants with MeJA, all the selected *SlJAZ* genes except *SlJAZ16* tended to increase. In particular, expression of the *SlJAZ18* gene increased 220.25-fold 24 h after treatment (Appendix A). The change in *SlJAZ* gene expression in the SA-treated group was almost the same as that in the MeJA-treated group, and the expression of *SlJAZ* genes in the SA-treated group changed relatively slowly. Expression of the *SlJAZ14* and *SlJAZ16* genes tended to decrease, while that of other *SlJAZ* genes tended to increase (Appendix A). Taken together, these results suggest that JA and SA may regulate the expression of *SlJAZ* genes.

### 2.8. Effects of SlJAZ25 Silencing on the S. lycopersici Defense Response

After the results of RNA-seq and qRT-PCR were combined, the *SlJAZ25* gene was selected as the experimental object for gene silencing to ultimately explore the role of *SlJAZ* genes in tomato resistance against *S. lycopersici* infection. With respect to tomato plants infected with *A. tumefaciens*, photobleaching was observed on the leaves of plants inoculated with TRV2-*PDS* at 20 dpi (Figure 9A). Expression of *SlJAZ25* in the leaves of TRV2-*SlJAZ25* plants was significantly lower than that in the leaves of TRV2-00 plants (Figure 9B). These results showed that the expression of *SlJAZ25* was silenced successfully in TRV2-*SlJAZ25* plants and that these plants could therefore be used for further research. Once photobleaching was observed in TRV2-*PDS* plants, the TRV2-00 plants and TRV2-*SlJAZ25* plants were inoculated with *S. lycopersici* simultaneously. At 3 dpi, severe damage or even perforation was observed on the leaves of the TRV2-00 plants, the edges of which were dark brown with yellow halos (Figure 9C). However, the TRV2-*SlJAZ25* plants grew normally, had dark green leaves, and showed no obvious spots or damage (Figure 9C).

### 2.9. Analysis of the Expression Patterns of Pathway Genes Related to SlJAZ25

To explore the possible molecular mechanism underlying the increased resistance to *S. lycopersici* in tomato plants with *SlJAZ25* gene silencing, the expression levels of the *SlJAR1*, *SlCOI1*, *SlJAZ25* and *SlMYC2* genes in the different treatment groups were analyzed. There was no significant difference in the expression of these four genes between the plants injected without *A. tumefaciens* and the TRV2-00 plants, indicating that experimental error caused by *A. tumefaciens* could be excluded. In the TRV2-*SlJAZ25* plants, expression levels of the *SlJAR1*, *SlCOI1* and *SlMYC2* genes increased by 2.78-, 3.45- and 2.10-fold, respectively (Figure 10). Three days after infection with *S. lycopersici* and 24 h after MeJA and SA treatment, the expression levels of all four genes increased significantly. Compared with that in the plants subjected to the other two treatments, expression of the four genes in the plants treated with MeJA increased most significantly.

### 2.10. Analysis of the Subcellular Localization of SlJAZ25

To determine the subcellular localization of the *SlJAZ25* gene product in *Nicotiana benthamiana*, a pCAMBIA1300s-*SlJAZ25*-GFP fusion expression vector was constructed and transformed into *Agrobacterium tumefaciens* GV1301 for infection into tobacco plants. As shown in Figure 11, pCAMBIA1300s-GFP (empty vector) displayed strong fluorescent signals in the cell membrane and nucleus of tobacco epidermal cells, but pCAMBIA1300s-*SlJAZ25*-GFP was detected only in the nucleus, which confirmed that *SlJAZ25* was expressed in the nucleus.

### 2.11. Analysis of the SlJAZ Gene Co-Expression Network and Yeast One-Hybrid System

The *SlJAZ* gene interaction network constructed by the STRING website revealed interactions among some *SlJAZ*s, among which there were many interactions between subgroups I and II but no interactions between genes in subgroup III and the other subgroups. In both subgroup IV and subgroup V, there were two genes that interacted with other *SlJAZ* genes (Figure 12A). The prediction results for the interactions between *SlJAZ25* and other genes showed that *SlJAZ25* interacted with 10 genes, including *ERF-C3*, *BSK2*, *COI1*, *SlWRKY45*, *SlWRKY46*, *PRO-P7*, and *MYC2* (Figure 12B). These genes have been shown to provide varying degrees of resistance to pathogens in other species. We speculate that *SlJAZ25* plays an important regulatory role in tomato resistance to gray leaf spots.

To verify the hypothesis that *SlJAZ25* can directly bind to the promoter sequence of *SlMYC2*, the *SlJAZ25* gene was cloned and inserted into a pGADT7 vector, and the 2000-bp upstream promoter sequence of the *SlMYC2* gene was cloned and inserted into a pHIS2 vector. The recombinant vector was subsequently introduced into yeast competent cells. As expected, the results showed that the yeast at all three concentrations grew normally on SD/-Leu/-Trp media, and that only *SlJAZ25*/*SlMYC2* Pro caused normal yeast growth on SD/-Leu/-Trp/-His media (supplemented with 3-AT; optimum concentration of 45 mM) (Figure 12C). Based on the experimental results, we concluded that there is a direct regulatory relationship between *SlJAZ25* and *SlMYC2*.

## 3. Discussion

### 3.1. Effect of the JA Pathway on Plant Disease Resistance

JA and its derivatives are collectively known as jasmonates and are plant hormones that are widely involved in plant growth and development, secondary metabolism, biotic and abiotic stresses, and other biological processes [45]. Although years of research and an accumulation of data have led to a certain understanding of the plant disease resistance signaling network involving JA, many uncertainties remain. After a pathogen invades a plant, the plant protects itself from damage through a series of physiological reactions, such as increasing the thickness of the cell wall and callose, secreting antibacterial substances, working synergistically with regulatory plant hormones, and stimulating programmed cell death [46]. A great deal of evidence shows that JA, SA, and ethylene (ETH) are the main signals that induce plant defense responses [47,48]. The JA pathway mediates the long-term systematic defense response (induced systemic resistance (ISR)) to resist the invasion of pathogens [49]. In the plant defense signaling network, there is a close relationship between the JA signaling pathway and the SA and ETH signaling pathways [50]. Moreover, JAZ proteins play an important biological role in regulating biotic and abiotic stresses in plants.

### 3.2. Distribution of the JAZ Gene Family in Plants

*JAZ* genes have been identified in a variety of plant species. There are 12 JAZs in *A. thaliana* [20], 15 in rice [15], 16 in maize [17], 30 in cotton [51], and 11 in grape [52]. We used tomato genomic data and predicted that 26 *SlJAZ* genes contain the conserved JAZ domain, which is approximately twice the number of *JAZ* genes present in *A. thaliana* and may be due to two genome-wide duplication events that occurred during tomato evolution [53]. In this study, the 26 *JAZ* genes were divided into five groups based on the conserved domains of the genes, which was consistent with the classification of the *JAZ* family members in *A. thaliana*. Genes in the same clade may have similar or complementary physiological functions.

### 3.3. Classification and Structure of JAZ Genes in Tomato

In plant evolutionary relationships, genes with similar intron–exon structures and conserved motif arrangements usually have similar functions [54]. The members of the *JAZ* families in *A. thaliana*, soybean, and other plant species are divided into five groups, and genes on the same branch have a similar structure and function [55]. In this study, the 26 *SlJAZ* genes were divided into five clades, which is consistent with the findings of previous studies. Most *SlJAZ* genes contain introns and exons, but the genes in clade III do not contain introns, indicating that they have been deleted from these genes. Studies have shown that precise loss and gain of introns may be important factors promoting the development of new genes [56]. According to the gene motif analysis, 13 motifs were identified among the 26 *SlJAZ* genes. Typical conserved JAZ motifs—motif 1 (TIFY motif) and motif 2 (Jas motif)—were present in all *SlJAZ* genes. Within the same clade, some motifs are unique and are the basis of gene family classification and functional differentiation.

### 3.4. The Promoters of SlJAZ Genes Contain a Large Number of Cis-Acting Elements Related to Disease Resistance

The specific binding of cis-acting elements and TFs in the promoter for regulating gene expression is not only the most important method for biological signal transduction, but also an important means by which genes can cooperate with other genes [57]. Many cis-acting elements have been found in plant pathogenesis-related (PR) gene promoters and are related to disease resistance, such as the ERF TF-specific GCC-box element and WRKY TF-binding W-box element [58,59]. In addition, a large number of hormone-related cis-acting elements have been identified [60]. To better understand the role of *SlJAZ* genes in tomato disease resistance, we analyzed the type, number, and distribution of cis-acting elements in the promoter regions of 26 *SlJAZ* genes. These cis-acting elements are divided into four categories according to their predicted functions: light-responsive elements, hormone-responsive elements, environmental stress-related elements, and development-related elements. The CGTCA motif, a JA-responsive element, is present in the promoter regions of most *SlJAZ* genes. In addition, there are a large number of other hormone-related cis-acting elements, such as ABREs, P-boxes, TCA elements and TGA elements. The presence of these cis-acting elements may be an important factor in the involvement of *SlJAZ*s in the JA-regulated disease resistance pathway.

### 3.5. Intraspecific and Interspecific Relationship of SlJAZ Genes

Gene duplication events constitute an important way for plants to evolve and for gene families to expand [61]. Studies have shown that 70% to 80% of angiosperms have experienced a gene duplication or polyploid event [62]. From the perspective of biological evolution, fragment replication, tandem duplication and translocation are effective means for generating new genes and developing resistance against foreign invaders [63]. There were 5 pairs of duplicated genes among the 26 *SlJAZ* genes in tomato, which indicates that tandem duplication events have greatly promoted the expansion of the *SlJAZ* gene family. To explore the phylogenetic relationships between tomato *SlJAZ* genes and those in other plant species, collinear relationships between tomato and *Arabidopsis thaliana* and potato were determined. Finally, 13 pairs of collinear *JAZ* genes between tomato and *A. thaliana* and 23 pairs of collinear *JAZ* genes between tomato and potato were identified. The number of homologous events between tomato and potato was much larger than that between tomato and *A. thaliana*, which is consistent with the small evolutionary distance between tomato and potato.

### 3.6. Transcriptome Combined with qRT-PCR Techniques to Reveal the Expression Profile of SlJAZ Genes

In addition, spatiotemporal expression analysis of the *SlJAZ* genes showed that different *SlJAZ* genes exhibited obvious tissue specificity, which provided a basis for these genes to participate in plant systematic physiological and biochemical activities. The expression of *SlJAZ2* and *SlJAZ3* in the tested tissues was very low, and the expression of *SlJAZ2* and *SlJAZ3* was not found in the transcriptomic data related to tomato resistance to gray leaf spots. The RNA-seq results were consistent with the qRT-PCR results, verifying the accuracy of the former. These plants were subsequently treated with MeJA and SA hormones, and the expression of almost all the *SlJAZ* genes was regulated by the two hormones. We therefore speculate that the pathways of these genes involved in plant disease resistance may be closely related to hormone regulation.

### 3.7. Resistance of the SlJAZ25 Gene to Gray Leaf Spots in Tomato and Its Hypothetical Pattern

In previous studies, *JAZ* genes have been shown to play an important role in responses to biotic and abiotic stresses [64]. In a study of tomato resistance to *Botrytis cinerea* regulated by the JA pathway, JAZs were shown to regulate the binding of *MYC2* to a group of bHLH proteins (*MTB1*, *MTB2* and *MTB3*) to regulate the termination of JA signaling [65,66]. The tomato gray leaf spots pathogen *S. lycopersici* is similar to *B. cinerea* and is a necrotrophic pathogen. We speculate that the regulation of tomato resistance to gray leaf spots is similar to the regulation mentioned above. Our results show that decreased expression of the *SlJAZ25* gene increased the resistance of tomato to gray leaf spots, and compared with that of TRV2-00 plants, the growth potential of TRV2-*SlJAZ25* plants significantly increased, indicating that *SlJAZ25* might also be related to growth and development. By combining our results with those of previous studies, we found that *SlJAZ25* interacts with *SlJAR1*, *SlCOI1*, and *SlMYC2*. After analyzing the qRT-PCR results, we speculate that there is a pattern in the process of tomato resistance to gray leaf spots, which is shown in Figure 13. Yeast hybridization experiments verified part of our hypothesis; that *SlJAZ25* could directly interact with the promoter region of *SlMYC2*. The role of a gene is not determined solely by the gene alone; the co-expression of a large number of genes in constructed network regulatory modules can aid in determining the ultimate effect [67]. There are a large number of co-expression and regulatory relationships among *SlJAZ* genes. These genes are related to each other, transducing invasion signals to all parts of the cell and promoting the cell to initiate the immune response.

### 3.8. Synergism between JA and Other Hormones

In the regulatory network of *JAZ* gene family members, JAZ proteins constitute not only an important component of JA signaling but also play an important role in the interaction between JA signals and other hormone signals [68,69]. *JAZ*s can interact with DELLA (highly conserved N-terminal DELLA motif) proteins to indirectly regulate gibberellin (GA) signaling, and DELLA proteins are important repressor proteins in the GA signal transduction pathway [70]. The degradation of JAZ proteins is related to the SA receptors *NPR3* and *NPR4*, which can be used to induce the expression of early JA-responsive genes and the synthesis of JA [71]. In addition, JAZ proteins have also been reported to interact with ETH, heteroauxin (IAA), and abscisic acid (ABA) [72]. In summary, JAZs play an important role in hormone signal transduction, and this study is helpful for understanding the mechanisms underlying the involvement of plant hormone signaling in plant growth and development and disease resistance.

## 4. Materials and Methods

### 4.1. Plant Growth and Treatments

The resistant tomato variety Motelle was obtained from the Institute of Vegetables and Flowers of the Chinese Academy of Agricultural Sciences (IVF·CAAS); the susceptible tomato variety Moneymaker and the common cultivated tomato variety Micro-Tom were maintained in our laboratory. Tomato seedlings were grown on sterilized nutrient soil in growth chambers. The environmental conditions were maintained as follows: during the day, there were 16 h of light (with a white light intensity of 200 μmol photons/m^2^/s), a temperature of 28 °C, and a relative humidity of 60%; during the night, there were 8 h of darkness, a temperature of 20 °C, and a relative humidity of 50%.

At 4 weeks of age, the plants were inoculated with a spore suspension of *S. lycopersici* (1 × 10^4^ spores/mL water). At 0 and 3 days post inoculation (dpi), the leaves of resistant and susceptible plants were collected for RNA-seq. The leaves of Micro-Tom plants were also collected to determine changes in gene expression at 0, 6, 12, 24, 36, and 48 h after inoculation. Ten plants per group were used. At 4 weeks of age, the Micro-Tom plants were evenly sprayed with solutions of 500 μM SA and 50 μM MeJA (Sigma-Aldrich, Saint Louis, MI, USA) individually. At 0, 6, 12, and 24 h after treatment, the leaves were collected to determine changes in gene expression. Each treatment group contained 10 plants, and all treatment groups included three biological replications.

### 4.2. Identification of JAZ Family Members in Tomato

Hidden Markov model (HMM) profile data for the TIFY (PF06200) and Jas (PF09425) JAZ domains were downloaded from the Pfam (protein domain family) database (http://pfam.xfam.org/, accessed 15 February 2021). Tomato genome and *A. thaliana* genome sequence data were obtained from the Ensembl Plants Database (http://plants.ensembl.org/index.html, accessed 17 February 2021). To identify *JAZ* genes in tomato and *A. thaliana*, all gene sequences were retrieved via HMMER 3.0 with default parameters and a cutoff of 0.01. Selection of *JAZ* genes in *A. thaliana* was performed based on a combination of previously reported results and our identification results. To further filter and verify tomato *JAZ* genes, all *SlJAZ*s were confirmed by the SMART database and CDD online tools. Genes with sequences that did not contain the TIFY or Jas domain sequences were excluded. To eliminate pseudogenes, REtrotransposed Gene EXPlorer was used for filtration. The physicochemical properties of these selected *SlJAZ*s, including their molecular weight, instability index value, and isoelectric point (pI), were predicted with ExPASy. The PSORT online tool was subsequently used to predict the subcellular localization of these proteins (http://psort1.hgc.jp/form.html, accessed 25 March 2021).

### 4.3. Chromosomal Mapping, Phylogenetic Relationship Analysis, and Gene Sequence Analysis of JAZ Gene Family Members in Tomato

*SlJAZ* gene chromosomal location data were retrieved from the Solanaceae Genomics Network (SGN) (https://solgenomics.net/, accessed 17 February 2021). A specific chromosome distribution map of *SlJAZ* genes was constructed with MG2C (map gene 2 Chromosome) online software (http://mg2c.iask.in/mg2c_v2.0/, accessed 28 March 2021). A phylogenetic tree of the *JAZ* sequences of *A. thaliana* and tomato was then constructed using MEGA X software. First, multiple sequence alignment was carried out by ClustalW, and default parameters were used. The multiple alignment results were subsequently used to construct a phylogenetic tree based on the NJ method with the Poisson model, homogeneous patterns, and pairwise deletions, with 1000 bootstrap replicates. Finally, the constructed evolutionary tree was optimized with the Evolview (Evolution view) online tool (https://www.evolgenius.info/evolview/, accessed 2 April 2021). The *SlJAZ* gene structure was determined using the Gene Structure Display Server 2.0 online program (http://gsds.cbi.pku.edu.cn/, accessed 4 April 2021). The protein sequences were analyzed with the MEME (multi EM for motif elicitation) online program (http:/meme.nbcr.net/meme/intro.html, accessed 8 April 2021), with the analysis parameter set to the default value. A phylogenetic tree (with clustering) was then constructed according to the results of the gene structure analysis.

### 4.4. Analysis of Cis-Acting Elements in the SlJAZ Gene Promoter

The promoter sequences of the *SlJAZ* genes (2 kb of the 5’ regulatory region upstream of the translation start sites) were obtained by searching the NCBI (National Center for Biotechnology Information) database (https://www.ncbi.nlm.nih.gov/, accessed 10 April 2021). PlantCARE (plant cis-acting regulatory element database) (http://bioinformatics.psb.ugent.be/webtools/plantcare/html/, accessed 10 April 2021) was used to predict the cis-acting elements, and TBtools (Toolkit for Biologists integrating various big-data handling tools) software was used to visualize their distribution (http://cj-chen.github.io/tbtools/, accessed 12 April 2021). An in-house Python program was used to quantify and classify the cis-acting elements.

### 4.5. Gene Duplication and Homeology Analysis of SlJAZ Genes

Genomic data for tomato, *A. thaliana* and potato were downloaded from the Ensembl Plants Database. Multiple collinear scanning toolkits (MCScanX) with default parameters were used to analyze gene duplication events (http://chibba.pgml.uga.edu/mcscan2/, accessed 16 April 2021). The syntenic relationship between orthologous *SlJAZ* genes of tomato and other selected species was determined using the Dual Synteny Plotter tool in TBtools software.

### 4.6. Analysis of SlJAZ Gene Expression Patterns via Transcriptomic Data

The transcriptomic data for buds, flowers, leaves, roots, and fruits of the tomato variety Heinz 1706 were downloaded from the Tomato Functional Genomic database (http://ted.bti.cornell.edu/cgibin/TFGD/digital/expression.cgi, accessed 21 April 2021). To measure gene expression levels, the total number of FPKM values of each gene was calculated based on the length of the gene and the count number of reads mapped to the gene. The FPKM values of *SlJAZ* genes in tomato-resistant and tomato-susceptible materials before and after inoculation with pathogens were obtained from the RNA-seq data (accession code: SRP097450). The experimental groups of resistant and susceptible plants sprayed with *S. lycopersici* conidial suspensions were named the RPI and SPI groups, respectively. The CK groups of resistant and susceptible plants sprayed with the same amount of distilled water were named the CK1 and CK2 groups, respectively. The FPKM means of three biological replicates of these genes were used to characterize their expression patterns after inoculation with *S. lycopersici*. The data were plotted after Z-score normalization across rows based on the mean FPKM values of each gene. The R package pheatmap was used to generate a heatmap.

### 4.7. VIGS Vector Construction and Agroinfiltration

Micro-Tom plants were used for VIGS infection. First, to avoid interference with the expression of other genes, we used the SGN VIGS tool to design specific fragments for silencing *SlJAZ25* (https://vigs.solgenomics.net/, accessed 26 April 2021) (Appendix A). Second, using Clone Express (CE) Design 1.04 software (https://crm.vazyme.com/cetool/singlefragment.html, accessed 26 April 2021), specific CE II primers were designed according to the *BamHI* restriction endonuclease digestion site of the pTRV2 vector (Miaoling Bio, Wuhan, China) and the selected *SlJAZ25* target fragment. The target fragment primer sequences used are listed in Appendix A, and the PCR profile was as follows: 95 °C for 3 min; 35 cycles of 95 °C for 15 s, 62 °C for 15 s, and 72 °C for 30 s; and 72 °C for 5 min. The target fragments were then ligated into a TRV2 empty vector. The constructed recombinant vector was transformed into competent *Escherichia coli* DH5α cells. Single clones were selected and cultured on LB medium supplemented with 50 μg/mL kanamycin and 100 μg/mL rifampicin. Samples of the bacterial liquid culture were sequenced to verify the vector recombination results. The TRV2-*PDS*, TRV2-00 and TRV2-*SlJAZ25* vector constructs were then introduced into *Agrobacterium tumefaciens* GV3101.

The *A. tumefaciens* isolates carrying TRV1 (from our laboratory), TRV2, TRV2-*PDS,* and TRV2-*SlJAZ25* were grown on LB medium supplemented with 50 μg/mL kanamycin and 50 μg/mL rifampicin. The recombinant colonies were transferred to LB liquid culture medium supplemented with the abovementioned antibiotics, after which they were allowed to grow to an OD_600_ of 0.25 at 28 °C with shaking at 200 rpm. *A. tumefaciens* was then concentrated by centrifugation and resuspended in induction medium buffer. *A. tumefaciens* cells containing TRV1 containing TRV2-derived constructs were mixed at a volumetric ratio of 1:1. Finally, the mixed bacterial liquid was infiltrated into the leaves of 4-week-old seedlings with a 1-mL syringe. Each treatment group comprised 10 plants, and the whole experiment was repeated three times.

### 4.8. Determination of Gene Silencing Efficiency and Phenotypic Observations

Once photobleaching was observed on the TRV2-*PDS* plants, the leaves of the TRV2-00 and TRV2-*SlJAZ25* plants were collected, and the relative gene expression was measured via qRT-PCR. Plants with no significant decrease in relative expression were excluded from further analysis. Three days after inoculation, the plant phenotypes were recorded to evaluate the disease.

### 4.9. Subcellular Localization of SlJAZ25

Construction and identification of the expression vector: Primers were designed according to the sequence of the pCAMBIA1300s-GFP plasmid and the full-length *SlJAZ25* coding DNA sequence (CDS), according to the specifications of the ClonExpress^®^ II One Step Cloning Kit, and *EcoRI* was selected as the site for linear plasmid digestion. The specific primers used are shown in Appendix A. The PCR amplification product was ligated into the pCAMBIA1300s-GFP vector using a ClonExpress^®^ II One Step Cloning Kit according to the manufacturer’s instructions. The recombinant plasmid was subsequently transformed into DH5α competent cells. The recombinant results were verified via PCR amplification, restriction endonuclease digestion, and sequencing. Positive clones were screened, and plasmids were extracted to obtain the pCAMBIA1300s-*SlJAZ25*-GFP vector construct. Instantaneous transformation of tobacco leaves and laser confocal microscopy observations: According to the method of Sparkes et al. [73], an *A. tumefaciens* suspension was injected into leaves of 4-week-old tobacco plants grown under normal growth conditions. After two days, the leaves injected with *A. tumefaciens* were removed to observe the green fluorescence of the GFP fusion protein using a Leica TCS SP2 AOBS spectral confocal microscope (Leica, Wetzlar, Germany).

### 4.10. Analysis of the Expression Network of SlJAZs in Tomato

An expression network of tomato *SlJAZ* genes was constructed with the STRING database (https://string-db.org, accessed 3 May 2021), the key *SlJAZ* genes expressed in response to tomato gray leaf spots pathogens were identified, and the genes interacting with *SlJAZ25* were selected through database resources. The basis of the interaction relationship derives from the experimental items, and other parameters are the default values. The output data were visualized with Cytoscape 3.8.2 (https://cytoscape.org/, accessed 5 May 2021).

### 4.11. Yeast One-Hybrid Assays

The complete CDS of the *SlJAZ25* gene and sequence of the 2000-bp upstream promoter region of the *SlMYC2* transcriptional initiation site were cloned. In accordance with the manufacturer’s requirements, a ClonExpress^®^ II One Step Cloning Kit (Vazyme, Nanjing, China) was used to clone the target fragment and ligate it into an expression vector. The full-length *SlJAZ25* CDS was amplified from tomato complementary DNA (cDNA) by PCR, cloned, and ligated into a pGADT7 vector. As described above, the 2-kb promoter fragment upstream of the initiation site of *SlMYC2* was isolated from tomato genomic DNA, amplified, and then cloned into a pHIS2 vector. The primer sequences used are listed in Appendix A. Both the pHIS2 bait vector and the pGADT7 prey vector were introduced into Y187 yeast (Code No. 630457, Takara, Shiga, Japan), which were subsequently cultured on SD/-Leu-Trp media (Takara, Shiga, Japan). After 3 days, the positive yeast strains were selected and diluted in sterilized distilled water to an OD_600_ of 0.1, and 2.5 μL of the suspension was spotted onto SD/-Leu-Trp-His media (Takara, Shiga, Japan) supplemented with 3-AT (at an optimum concentration of 45 mM). The plates were subsequently incubated for 3–7 days at 30 °C.

### 4.12. Total RNA Extraction, cDNA Synthesis and qRT-PCR

Total RNA of the tomato leaves was extracted according to the instructions provided with the MiniBEST Plant RNA Extraction Kit (TaKaRa, Shiga, Japan). A Transcript II One-Step gDNA Removal and cDNA Synthesis Kit (TransGen Biotech, Beijing, China) was used to synthesize cDNA. The cDNA samples obtained were stored at −80 °C until use.

The target sequences of the genes were amplified using specific PCR primers (Appendix A). The primers were designed with Primer Premier 5 software (http://www.premierbiosoft.com/primerdesign/, accessed 10 May 2021), and the sequences were sent to the Beijing Genomics Institute (BGI, Beijing, China) for synthesis. For gene expression analyses, qRT-PCR was performed for three independent biological replicates using AceQ^®^ qPCR SYBR^®^ Green Master Mix (Vazyme, Nanjing, China) in a 20-μL volume on a qTOWER^3^G Real-time System (Analytik Jena AG, Jena, Germany). *EF1α* was used as an internal control for normalization of the obtained data, and the relative expression was calculated using the 2^−ΔΔCT^ method.

## 5. Conclusions

Tomato gray leaf spots caused by *S. lycopersici* are a recent epidemic disease. After plants are infected by pathogens, a series of immune responses occur in the plant to resist continued pathogen invasion. As an important part of the JA signal transduction pathway, JAZs play a prominent role in plant disease resistance. First, we identified 26 *SlJAZ* genes in tomato for the first time. These genes clustered into five branches, and their gene structure and motif distribution in the same group were similar. The promoter regions of these genes contained many cis-acting elements related to disease resistance. The spatiotemporal expression characteristics of *SlJA*Z genes varied widely. Analysis of RNA-seq data showed various expression patterns of *SlJAZ*s after pathogen inoculation. Most *SlJAZ* genes could be induced by *S. lycopersici*, MeJA, and SA, but their expression patterns differed. We subsequently confirmed the negative regulatory effect of the *SlJAZ25* gene on tomato resistance to gray leaf spots. In addition, our results revealed that *SlJAZ25* targeted *SlMYC2* and that those other regulatory genes were involved in the regulatory module of tomato resistance to gray leaf spots. These results not only show the importance of *SlJAZ*s in tomato disease resistance but also lay a good foundation for further study of the potential regulatory mode of *SlJAZ* genes.

## Figures and Tables

**Figure 1 ijms-22-09974-f001:**
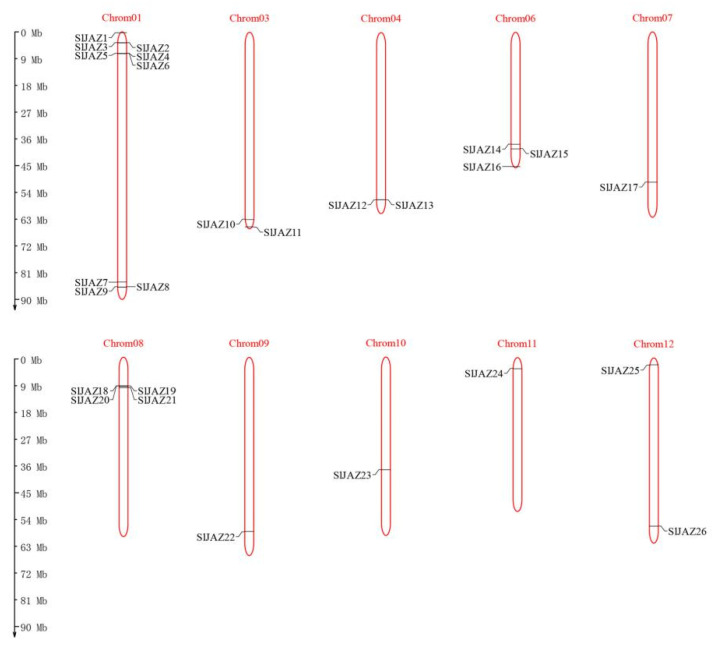
Chromosomal distribution of tomato *JAZ* genes (*SlJAZs*). Chromosomes without *SlJAZ* genes are not shown. The length of the chromosome can be estimated using the scale on the left.

**Figure 2 ijms-22-09974-f002:**
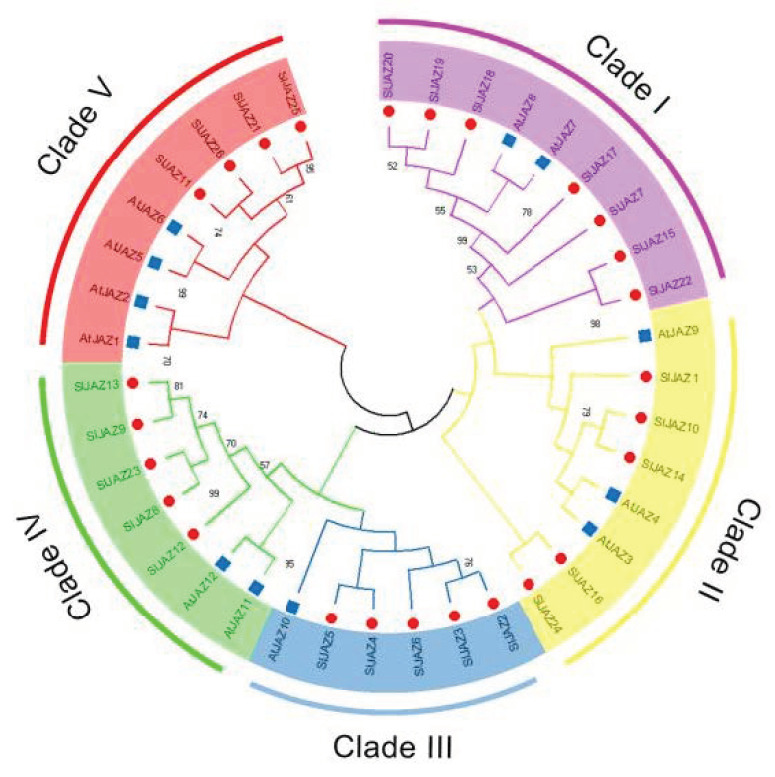
Phylogenetic relationships of 26 SlJAZs in tomato and 12 AtJAZ protein sequences in *A. thaliana*. Different branches are shown in different colors. Nodes with a bootstrap support value of less than 50 are not shown.

**Figure 3 ijms-22-09974-f003:**
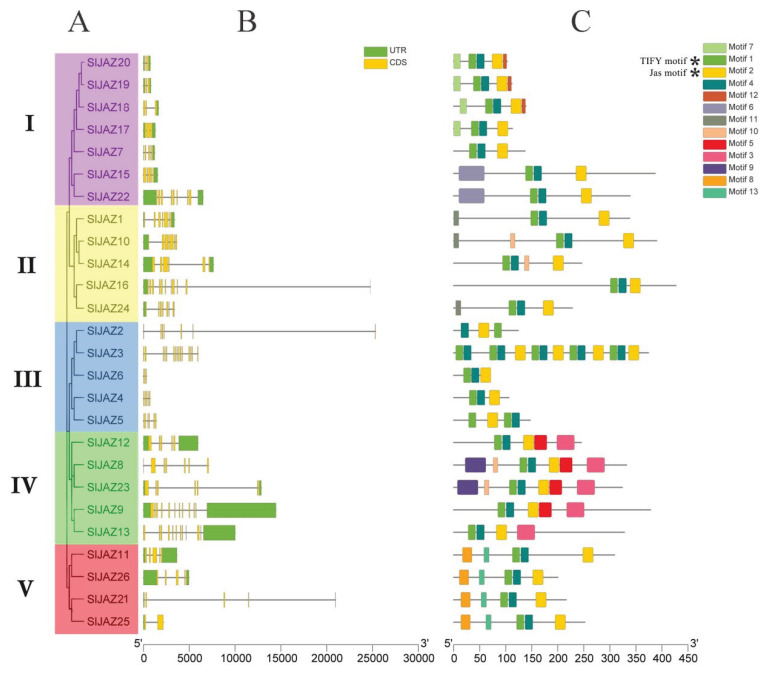
Gene structure of *JAZ* gene family members in tomato. (**A**) Phylogenetic tree of the *JAZ* gene family in tomato. Branches of the same color represent the same subgroup. (**B**) Distribution of introns and exons. (**C**) Distribution of motifs. Asterisks indicate the motifs shared by *SlJAZ* genes.

**Figure 4 ijms-22-09974-f004:**
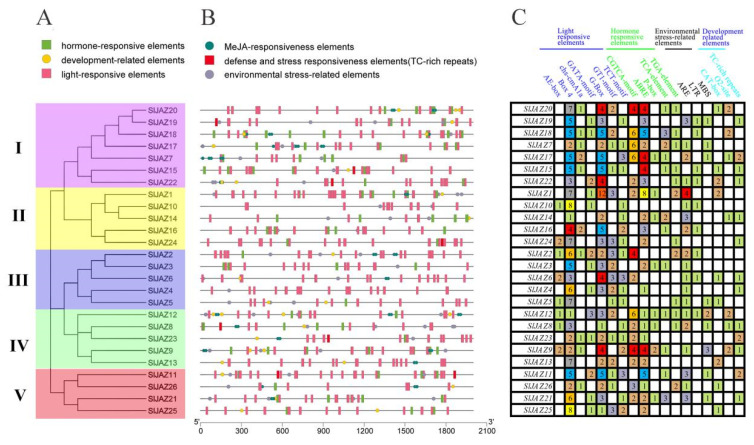
Distribution of cis-acting elements in the promoter regions of *SlJAZ* genes. (**A**) Phylogenetic tree of *JAZ* gene family members in tomato. The different background colors correspond to the different branches. (**B**) Distribution of cis-acting elements in the promoter region (−2000 bp of the initiation codon) of *SlJAZ* genes. The different colored blocks represent the different types of cis-acting elements and their locations in each *SlJAZ* gene. (**C**) Statistics for the 18 cis-acting elements in the *SlJAZ* gene. The 18 types of cis-acting elements were divided into four categories. The different colors and numbers in the grid indicate the numbers of different cis-acting elements in these *SlJAZ* genes.

**Figure 5 ijms-22-09974-f005:**
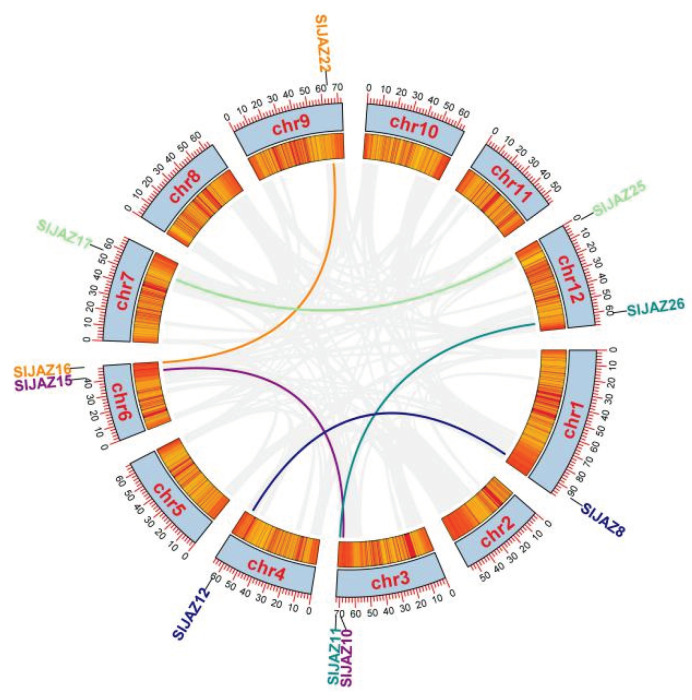
*SlJAZ* gene duplication events. The inner circle shows the distribution of genes on each chromosome. The gene names for each pair of duplication events are displayed in the same color.

**Figure 6 ijms-22-09974-f006:**
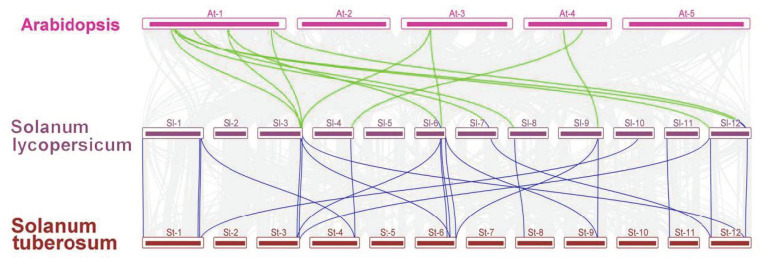
Syntenic relationships between homologous *SlJAZ* genes of tomato and other species. The *JAZ* gene pairs between different species are highlighted with different colored lines.

**Figure 7 ijms-22-09974-f007:**
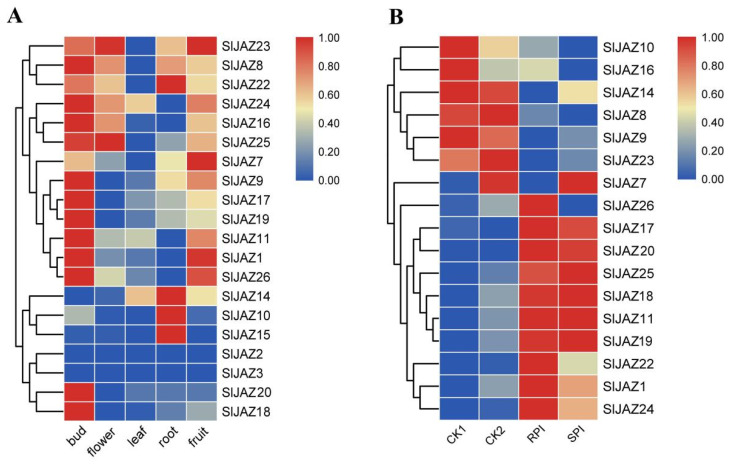
Expression profiles of *SlJAZ* genes in tomato. (**A**) Expression patterns of *SlJAZ* genes in different tomato tissues and organs. (**B**) Expression profiles of *SlJAZ* genes in plants inoculated with *S. lycopersici* and in CK plants. The relative transcript levels of *SlJAZ*s in the RNA-seq data were plotted after Z-score normalization was performed across the rows based on the absolute FPKM value of each gene. The colors from blue to red represent the range of the relative expression levels from low to high, respectively.

**Figure 8 ijms-22-09974-f008:**
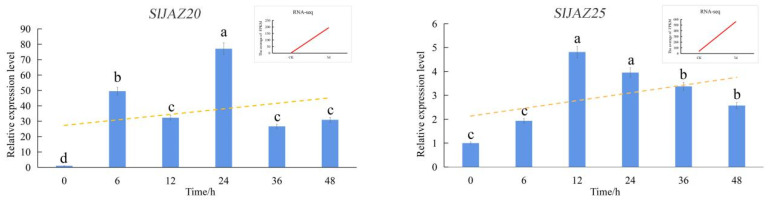
Analysis of the expression of the *SlJAZ20* and *SlJAZ25* genes in tomato after inoculation with *S. lycopersici*. The mean expression value was calculated from three independent replicates. Vertical bars show the standard deviations. Different letters indicate significant differences based on Duncan’s multiple range test (*p* ≤ 0.05).

**Figure 9 ijms-22-09974-f009:**
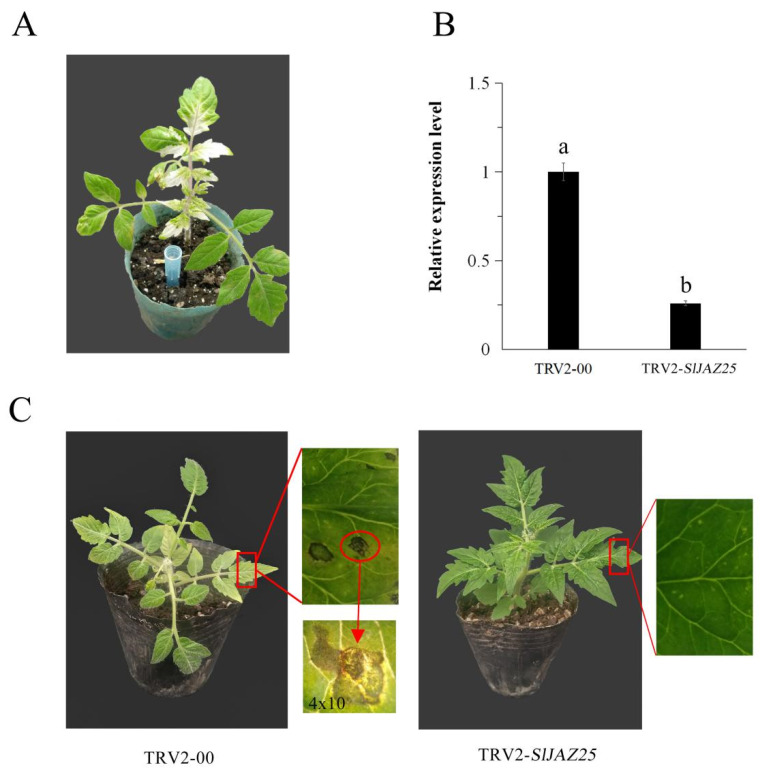
Determination of *SlJAZ25* gene silencing efficiency and phenotypes of TRV2-00 and TRV2-*SlJAZ25* plants at 3 dpi. (**A**) TRV2-*PDS* plants with a characteristic photobleaching phenotype at 20 dpi. (**B**) Relative expression levels of *SlJAZ25* in TRV2-00 and TRV2-*SlJAZ25* plants. (**C**) Phenotypes of TRV2-00 and TRV2-*SlJAZ25* plants at 3 dpi. The data presented are the means ± standard deviations of three independent experiments, and the different letters above the columns indicate significant differences at the *p* ≤ 0.05 level.

**Figure 10 ijms-22-09974-f010:**
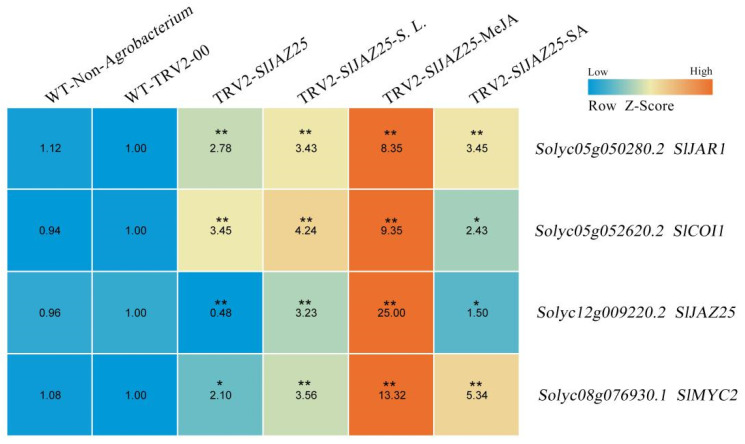
Expression of pathway genes related to *SlJAZ25* in the different treatment groups via whole-plant inoculation assays. The mean expression value was calculated from the results of three independent replicates. The number in the boxes indicates the rate of gene expression. A table was plotted after Z-score normalization across each row based on the gene expression data from each gene in the different treatment groups. The colors vary from blue to red, representing the scale of the relative expression levels. Asterisks indicate significant differences (*, *p* ≤ 0.05; **, *p* ≤ 0.01) based on Duncan’s multiple range test.

**Figure 11 ijms-22-09974-f011:**
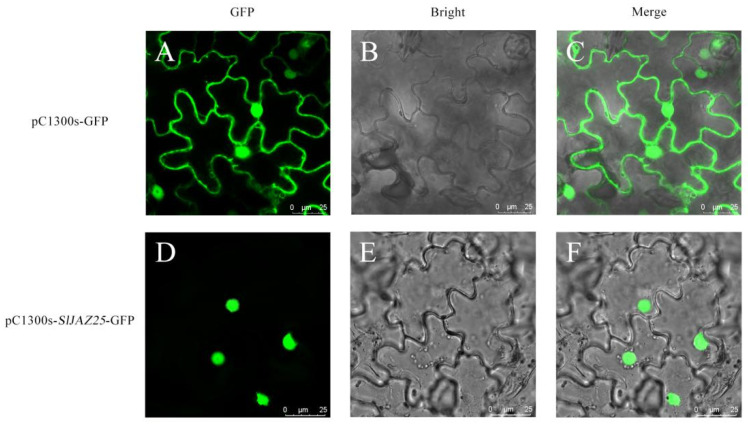
Fluorescence of *SlJAZ25* in tobacco leaf cells. (**A**–**C**) Dark field, bright field, and merged images of pCAMBIA1300s-GFP fluorescence. (**D**–**F**) Dark field, bright field, and merged images of pCAMBIA1300s-*SlJAZ25*-GFP fluorescence. Scale bars = 25 µm.

**Figure 12 ijms-22-09974-f012:**
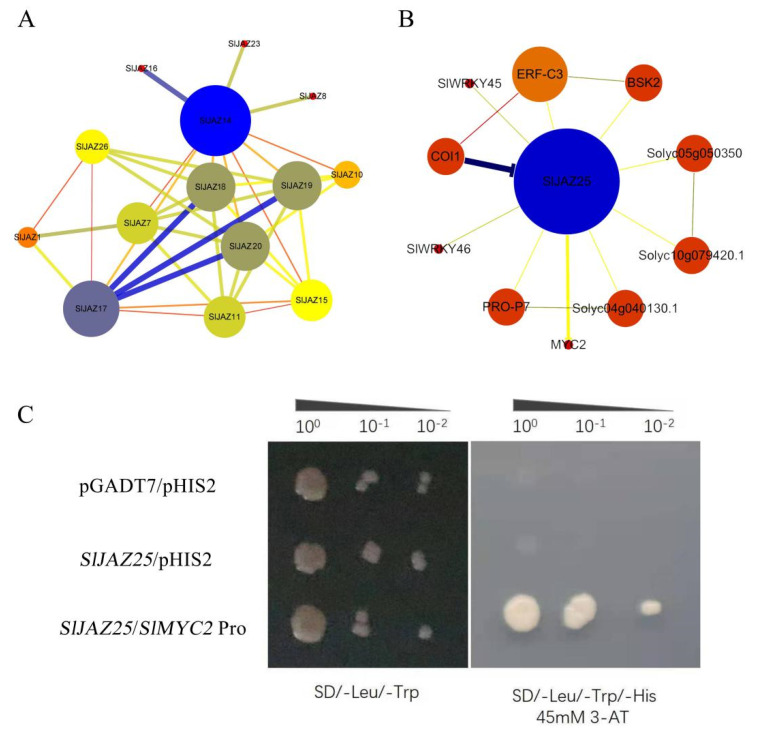
Co-expression network of *SlJAZ*s and the regulatory network relationship of *SlJAZ25* in tomato. *SlJAZ25* binds to the *SlMYC2* promoter. (**A**) Co-expression network of *SlJAZ*s. (**B**) Regulatory network relationships of *SlJAZ25* in tomato. (**C**) Results of a yeast one-hybrid experiment comparing the tomato *SlJAZ25* gene and the *SlMYC2* promoter.

**Figure 13 ijms-22-09974-f013:**
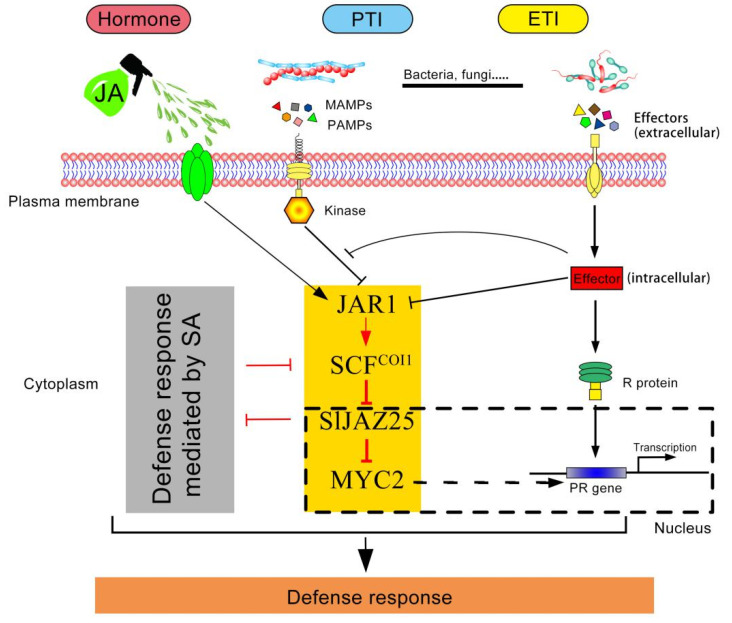
Hypothetical pattern of the immune response involving the participation of *SlJAZ25* in tomato.

## Data Availability

The raw sequencing data of this article are stored in the NCBI Sequence Read Archive under accession number SRP097450.

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
