# Peer review of "Genome-Wide Identification, Characterization and Expression Analysis of the JAZ Gene Family in Resistance to Gray Leaf Spots in Tomato"

_ijms, 2021, doi:10.3390/ijms22189974_

Round 1

Reviewer 1 Report

General observations

In a bioinformatics approach, the authors identified 26 members of the JAZ gene family and analyzed their physical and chemical properties, their subcellular localization and gene structure. The study further elucidated the regulatory role of SlJAZ25 genes in the grey leaf spot resistance of tomatoes. A decrease in the expression of the SlJAZ25 gene resulted in increased resistance of tomato to grey leaf spot. This is a well-written manuscript with a sound scientific approach that merits publication in the International Journal of Molecular Sciences.

Some specific comments and suggestions for improvement are made below.

Specific comments/suggestions:

L3: the JAZ gene family in resistance to gray leaf spot

Please change 'gray leaf spot' to 'grey leaf spot' throughout the text.

L57-58: However, there are few studies on resistance to gray leaf spot.

This sentence should be deleted because it is a duplication of the next, slightly longer sentence

L71-72: mutants are more susceptible to Pseudomonas syringae tomato DC3000, and this may due to the accumulation of coronatine [27].

Grammar: the word ‘be’ is missing after ‘may’.

L78: these results suggest that JAZ proteins may have a negatively regulate plant resistance,

The following part of the sentence needs rewording to become clear to readers: “may have a negatively regulate plant resistance”

L84: causing the disease was determined to S. lycopersici [32].

Please insert the word ‘be’ after ‘to’: was determined to be S. lyopersici.

L87-88: Resistance against gray leaf spot has been identified in the wild species Solanum pimpinellifolium and provided by a single incompletely dominant gene

Please insert the word ‘is’ before ‘provided’.

L135-139: Please check and correct font size!

L356-365: Please check and correct the font size of this paragraph!

L361-362: and may be due to two genome-wide duplications events that occurred during tomato evolution.

This statement requires a supporting reference.

L415: . These plants were subsequently were treated with MeJA

Duplication of the word ‘were’. Please remove the second one.

L444-445: and DELLAs protein are important repressor proteins in the GA signal transduction pathway [60].

This should read: and DELLA proteins are important…

L598: was used to clone the target fragment and ligated it into an expression vector

Suggested edit: was used to clone the target fragment and ligate it into an expression vector

L631: The spatiotemporal expression characteristics of SlJAZ genes are vary widely.

Grammar! Please remove the word ‘are’ in this sentence.

Author Response

Response to Reviewer 1 Comments

Thank you for your comments concerning our manuscript entitled “Genome-wide Identification, Characterization and Expression Analysis of the JAZ Gene Family in Resistance to Grey Leaf Spots in Tomato” (Manuscript ID: ijms-1340821). Those comments are all valuable and very helpful for revising and improving our paper, as well as the important guiding significance to our researches. We have studied comments carefully and have made correction which we hope meet with approval. Revised portion are marked in red in the the revised manuscript. Responds to all comments in this article are as follows:

Point 1: L3: the JAZ gene family in resistance to gray leaf spot.

Please change 'gray leaf spot' to 'grey leaf spot' throughout the text.

Response 1: Thank you very much for your suggestions.

Line 3, 21, 23, 26, 30, 31, 65, 89, 96, 103, 119, 350, 446, 451, 456, 458, 459, 464, 631, 668, 679 and 681: We have made correct changes and examined them carefully in the full text. Meanwhile, according to the suggestion of reviewer 3, we replaced all 'gray leaf spot' in the manuscript with 'grey leaf spots'.

Point 2: L57-58: However, there are few studies on resistance to gray leaf spot.

This sentence should be deleted because it is a duplication of the next, slightly longer sentence.

Response 2: Thank you very much for your responsible comments.

Line 64: This sentence has been deleted.

Point 3: L71-72: mutants are more susceptible to Pseudomonas syringae tomato DC3000, and this may due to the accumulation of coronatine [27].

Grammar: the word ‘be’ is missing after ‘may’.

Response 3:  Thank you for your careful correction.

Line 79: This point has been correctly modified and carefully checked in the full text.

Point 4: L78: these results suggest that JAZ proteins may have a negatively regulate plant resistance.

The following part of the sentence needs rewording to become clear to readers: ‘may have a negatively regulate plant resistance’.

Response 4: Thank you very much for your kindly advice.

Line 86-87: To make the reader understand more clearly, the sentence has been replaced with ‘these results suggest that the expression of JAZ protein in plants may reduce plant defense against diseases’.

Point 5: L84: causing the disease was determined to S. lycopersici [32].

Please insert the word ‘be’ after ‘to’: was determined to be S. lyopersici.

Response 5: Thank you very much for your responsible comments. 

Line 93: This point has been correctly modified, and carefully checked in the full text.

Point 6: L87-88: Resistance against gray leaf spot has been identified in the wild species Solanum pimpinellifolium and provided by a single incompletely dominant gene.

Please insert the word ‘is’ before ‘provided’.

Response 6: Thank you for your careful inspection.

Line 97: This point has been correctly modified and carefully checked in the revised manuscript.

Point 7: L135-139: Please check and correct font size!

Response 7: Thank you very much for your suggestions.

Line 153-158: According to your suggestion, this part has been corrected correctly.

Point 8: L356-365: Please check and correct the font size of this paragraph!

Response 8: Thank you very much for your responsible comments.

Line 382-390: This part has been corrected correctly and carefully checked in the full text.

Point 9: L361-362: and may be due to two genome-wide duplications events that occurred during tomato evolution.

This statement requires a supporting reference.

Response 9: Thank you for your constructive comments.

Line 387, 825-828: To enable readers to understand the article clearly, a reference has been added, which is as follows:

[53] Lin, T.; Zhu, G.; Zhang, J.; Xu, X.; Yu, Q.; Zheng, Z.; Zhang, Z.; Lun, Y.; Li, S.; Wang, X.; Huang, Z.; Li, J.; Zhang, C.; Wang, T.; Zhang, Y.; Wang, A.; Zhang, Y.; Lin, K.; Li, C.; Xiong, G.; Xue, Y.; Mazzucato, A.; Causse, M.; Fei, Z.; Giovannoni, J.J.; Chetelat, R.T.; Zamir, D.; Städler, T.; Li, J.; Ye, Z.; Du, Y.; Huang, S. Genomic analyses provide insights into the history of tomato breeding. Nat Genet. 201446, 1220-1226, doi: 10.1038/ng.3117.

Point 10: L415: These plants were subsequently were treated with MeJA.

Duplication of the word ‘were’. Please remove the second one.

Response 10: Thank you very much for your careful check. I am sorry for our carelessness.

Line 447-448: This point has been revised correctly and checked carefully in the full text.

Point 11: L444-445: and DELLAs protein are important repressor proteins in the GA signal transduction pathway [60].

This should read: and DELLA proteins are important…

Response 11: Thank you for your careful inspection.

Line 480: This point has been revised correctly and checked carefully in the full text.

Point 12: L598: was used to clone the target fragment and ligated it into an expression vector.

Suggested edit: was used to clone the target fragment and ligate it into an expression vector.

Response 12: Thank you very much for your responsible comments.

Line 640-641: According to your constructive suggestion, this sentence has been replaced with ‘was used to clone the target fragment and ligate it into an expression vector’.

Point 13: L631: The spatiotemporal expression characteristics of SlJAZ genes are vary widely.

Grammar! Please remove the word ‘are’ in this sentence.

Response 13: Thank you for your careful correction.

Line 675: This point has been revised correctly and checked carefully in the full text.

Thank you again for your comments and hope to learn more from you!

Reviewer 2 Report

In the submitted manuscript “Genome-wide identification, characterization and expression analysis of the JAZ gene family in resistance to gray leaf spot in tomato”, 26 members of the JAZ gene family were identified in tomato for the first time. The characteristics of the validated SlJAZ genes were investigated through phylogenetic, chromosomal location, collinearity, genetic structure, conserved protein domain, and cis-acting element analyses. The paper is timely and will be of interest to professionals currently working in this area. I feel that the manuscript is well-written, and it can be published in IJMS journal after addressing some concerns.

There are some comments about this submission:

Line 20: which hormone regulation? ABA? Auxin? Cytokinin? JA? Nitric oxide?

Line 32: Please change "various stresses, such as diseases, insect pests, drought, and cold, lead to a" to " various stresses (e.g., diseases, insect pests, drought, and cold) lead to a".

Lines 38-40: Please change this sentence. Because, as you know that, for a specific function, not only gene networks are important, but also there are other factors such as transcriptional and epigenetic regulations that play an important role in this matter.

Line 57-59: please change it to "Although there are some studies (NEED REFERENCES) on resistance to gray leaf spot, the role of the JAZ gene family in tomato resistance to gray leaf spot has been rarely studied (NEED REFERENCES)".

Lines 65-66: the authors mentioned that "previous studies" but they provided only one reference. Please provide more references.

Lines70: Please provide more reference(s).

Lines 135-139: Please keep consistency in using the font size.

Lines 154-155: Please replace figure 1 with a figure with better quality.

Lines 194-195: Please replace figure 2 with a figure with better quality.

Line 260: Please replace figure 4 with a figure with better quality.

Lines 356-365: Please keep consistency in using the font size.

Author Response

Response to Reviewer 2 Comments

Thank you very much for your attention ,evaluation and comments on our manuscript (Title: Genome-wide Identification, Characterization and Expression Analysis of the JAZ Gene Family in Resistance to Grey Leaf Spots in Tomato; Manuscript ID: ijms-1340821). We have revised the manuscript according to your kind advices and detailed suggestions. All the changes in this article have been marked in red font. we sincerely hope that you will approve the amendments to this manuscript. Responds to all comments in this article are as follows:

Point 1: Line 20: which hormone regulation? ABA? Auxin? Cytokinin? JA? Nitric oxide?

Response 1: Thank you very much for your suggestions.

Line 20: In order to enable readers to interpret the manuscript more clearly, we replace 'hormone regulation' with 'methyl jasmonate (MeJA) and salicylic acid (SA)'.

Point 2: Line 32: Please change "various stresses, such as diseases, insect pests, drought, and cold, lead to a" to " various stresses (e.g., diseases, insect pests, drought, and cold) lead to a".

Response 2: Thank you very much for your responsible comments.

Line 36-37: This part has been modified correctly.

Point 3: Lines 38-40: Please change this sentence. Because, as you know that, for a specific function, not only gene networks are important, but also there are other factors such as transcriptional and epigenetic regulations that play an important role in this matter.

Response 3: Thank you for your constructive suggestions.

Line 43-46, 710-714: According to your suggestion, this part has been appropriately supplemented and modified. The changed content is ‘Members of all kinds of gene families perform specific functions and connect to form a network that controls plant disease resistance; in addition, other factors, such as transcription and epigenetic regulation, also play an important role’. Two references have been added, as follows:

[5] Galiano, L.M.J.; Hernández, G.A.I.; Salvador, C.O.; Rausell, C.; Real, M.D.; Escamilla, M.; Camañes, G.; Agustín, G.P.; Bosch, G.C.; Robles, G.I. Epigenetic regulation of the expression of WRKY75 transcription factor in response to biotic and abiotic stresses in Solanaceae plants. Plant Cell Rep. 2018, 37, 167-176, doi: 10.1007/s00299-017-2219-8.

[6] Benoit, M.; Drost, H.G.; Catoni, M.; Gouil, Q.; Gomollon, L.S.; Baulcombe, D.; Paszkowski, J. Environmental and epigenetic regulation of Rider retrotransposons in tomato. PLoS Genet. 2019, 15, e1008370, doi: 10.1371/journal.pgen.1008370.

Point 4: Line 57-59: please change it to "Although there are some studies (NEED REFERENCES) on resistance to gray leaf spot, the role of the JAZ gene family in tomato resistance to gray leaf spot has been rarely studied (NEED REFERENCES)".

Response 4: Thank you very much for your kindly advice.

Line 64-66, 751-760: According to your suggestion, the sentence has been modified correctly. Four references have been added, as follows:

[23] Liu, J.; Wang, X. Early recognition of tomato gray leaf spot disease based on MobileNetv2-YOLOv3 model. Plant Methods. 2020, 16, 83, doi: 10.1186/s13007-020-00624-2.

[24] Sun, Y.; Wang, T.; Liu, M.; Nie, Z.; Yang, H.; Zhao T.; Xu, X.; Jiang, J.; Li, J. Virus-induced gene silencing of SlPKY1 attenuates defense responses against gray leaf spot in tomato. Scientia Horticulturae2020, 264, 109149, 10.1016/j.scienta.2019.109149.

[25] Medina, R.; Franco, M.E.E.; Cabral, L.D.; Bahima, J.V.; Patriarca, A.; Balatti, P.A.; Saparrat, M.C.N. The secondary metabolites profile of Stemphylium lycopersici, the causal agent of tomato grey leaf spot, is complex and includes host and non-host specific toxins. Australasian Plant Pathology. 2021, 50, 105-115, doi: 10.1007/s13313-020-00753-1.

[26] Saito, R.; Hayashi, K.; Nomoto, H.; Nakayama, M.; Takaoka, Y.; Saito, H.; Yamagami, S.; Muto, T.; Ueda, M. Extended JAZ degron sequence for plant hormone binding in jasmonate co-receptor of tomato SlCOI1-SlJAZ. Sci Rep. 2021, 11, 13612, doi: 10.1038/s41598-021-93067-1.

Point 5: Lines 65-66: the authors mentioned that "previous studies" but they provided only one reference. Please provide more references.

Response 5: Thank you very much for your careful check.

Line 72-74, 773-776: Based on your suggestion, two references have been added, as follows:

[32] Collins, J.; Grady, K.o.; Chen, S.; Gurley, W. The C-terminal WD40 repeats on the TOPLESS co-repressor function as a protein-protein interaction surface. Plant Mol Biol. 2019, 100, 47-58, doi: 10.1007/s11103-019-00842-w.

[33] Huang, P.Y.; Zhang, J.; Jiang, B.; Chan, C.; Yu, J.H.; Lu, Y.P.; Chung, K.; Zimmerli, L. NINJA-associated ERF19 negatively regulates Arabidopsis pattern-triggered immunity. J Exp Bot. 2019, 70, 1033-1047, doi: 10.1093/jxb/ery414.

Point 6: Lines70: Please provide more reference(s).

Response 6: Thank you very much for your responsible comments.

Line 76, 777-779: Based on your suggestion, a reference have been added, as follows:

[34] Monte, I.; Zorrilla, F.J.M.; Casado, G.G.; Zamarreño, A.M.; Mina, G.J.M.; Nishihama, R.; Kohchi, T.; Solano, R. A Single JAZ Repressor Controls the Jasmonate Pathway in Marchantia polymorpha. Mol Plant. 2019, 12, 185-198, doi: 10.1016/j.molp.2018.12.017.

Point 7: Lines 135-139: Please keep consistency in using the font size.

Response 7: Thank you for your careful correction.

Line 153-158: This part has been corrected correctly and carefully checked in the full text.

Point 8: Lines 154-155: Please replace figure 1 with a figure with better quality.

Response 8: Thank you for your careful inspection.

Line 149, 175 and 178: According to your suggestion and the comments of the reviewer 3, we split the original figure 1 into the current version of figure 1, figure 2, figure 3 and supplementary figure 1, and the clarity and quality of these figures have been significantly improved.

Point 9: Lines 194-195: Please replace figure 2 with a figure with better quality.

Response 9: Thank you for your constructive suggestions.

Line 198, 221 and 224: According to your suggestion and the comments of the reviewer 3, we split the original figure 2 into the current version of figure 4, figure 5 and figure 6, and the clarity and quality of these figures have been significantly improved.

Point 10: Line 260: Please replace figure 4 with a figure with better quality.

Response 10: Thank you very much for your responsible comments.

Line 283: Based on your suggestion and the comments of reviewer 3, we selected two representative charts from the original figure 4 and combined them side by side. More details can be shown, and the optimized original figure 4 has been uploaded as supplementary figure 2. The clarity and quality of these figures have been significantly improved.

Point 11: Lines 356-365: Please keep consistency in using the font size.

Response 11: Thank you for your careful correction.

Line 382-390: According to your suggestion, this part has been corrected correctly.

Thank you again for your comments and hope to learn more from you!

Reviewer 3 Report

In this manuscript, the authors did Genome-wide identification, characterization, and expression analysis of the JAZ gene family in resistance to gray leaf spots in tomato. Here author identified 26 JAZ genes in tomato. The physical and chemical properties predicted subcellular localization, gene structure, cis-acting elements, and interspecies collinearity of 26 SlJAZ genes were subsequently analyzed. RNA-seq data combined with qRT-PCR analysis data showed that most SlJAZ genes were induced in response to Stemphylium lycopersici, MeJA and SA. TRV2-SlJAZ25 plants were more resistant to tomato gray leaf spots than TRV2-00 plants were. It indicated that SlJAZ25 plays a negative regulatory role in tomato resistance to gray leaf spots. By combining the results of previous studies and those of this study, the author speculated that SlJAZ25 might be closely related to hormone regulation. SlJAZ25 interacts with SlJAR1, SlCOI1, SlMYC2, and other resistance-related genes to form a regulatory network, and these genes play an important role in regulating tomato gray leaf spots. The subcellular localization results showed that the SlJAZ25 gene was located in the nucleus. According to the authors, this study is the first to identify and analyze JAZ family genes in tomato via bioinformatics approaches, which further clarified the regulatory role of SlJAZ25 genes in tomato resistance to gray leaf spots and provided new ideas for improving plant disease resistance.

Data is solid in this study, but the manuscript needs serious refinement to consider further for publication. I have given my comments in detail below.

  1. Figure 1 is strange. The author tried to adjust 3 figures into one; hence the font letter seems more prominent than the real figure. Also, After naming as A or B, the subfigure is named as a and b, which looks odd. My suggestion is to separate all figures and then just name them A, B, and C.
  2. The same problem happens in figure 2. Please separate all figures. In Figure 2B, font B looks bigger than the figure itself.
  3. Figure 4 is very small. Gene's name looks more prominent than the graph itself. My suggestion is to make figure potraite instead of landscape ans add only 2 genes side by side. What is the control in this figure. Make all control as 1 or 0. It should be the same.
  4. Consider dividing the discussion part into subsections.
  5. Make changes at

L11 in the response to TO in response to.

L12 What is JAZ? Write down the full form.

L17 MeJA and SA. TRV2 full form?

L20 may be closely to might be closely.

L17, L19, L22, L25, L27, L58, and L326, L635, L637 leaf spot to leaf spots.

L33 continually at-tacked to continually attacked.

At L35 it is written anti-bacterial and at L347 its is antibacterial. Please be consistant.

L36 hormones and so on are all? This line is strange.

L39 and connect with each other to form a network to and connect to form a network.

L40 further eluci-dating to further elucidating.

L42 ZIM full form?

L62 TIFY full form? At L42 ZIM domain is JAZ however, at L62 ZIM domain is TIFY! Which is true?

L71 this may due to this may be due.

L78 may have a negatively regulate to may have negatively regulated.

L80 most serious plant diseases to most severe plant diseases.

L105 significantly increases what?

L114-115 Full form and weblink of HMMER, SMART, and CDD, also write down the date when you have accessed them latest.

L124 Write down molecular weight instead of MW first time.

L134 in Section 2.2 for phylogenetic tree author must have used amino acid instead of nucleotides so write down all gene name nomal not italic as they are candidate proteins not nucleotide.

L209 What is FPKM and TFGD full form?

L245 On the basis of to Based on.

L259 the expression of SlJAZ genes may be regulated by JA and SA to JA and SA may regulate the expression of SlJAZ genes.

L301 boxes indicate to boxes indicates.

L440 Pattern of the immune to Hypothetical pattern of the immune.

L444 Full form of DELLA.

Write down full form of software or webtool you have used in material method secton along with the date when you have used those.

L631 SlJAZ genes are vary widely to SlJAZ genes vary widely.

L636 that other regulatory genes to that other regulatory gene or to those other regulatory genes.

Author Response

Response to Reviewer 3 Comments

Thank you very much for your constructive comments. Based on your comment and request, we have made extensive modification on the original manuscript. About the English writing of the manuscript, we ask for native English speaker to revise the paper in an all-round way. These combined figures has been split, and the clarity and quality have been greatly improved. All the changes in this article have been marked in red font. we sincerely hope that you will approve the amendments to this manuscript. Responds to all comments in this article are as follows:

Point 1: Figure 1 is strange. The author tried to adjust 3 figures into one; hence the font letter seems more prominent than the real figure. Also, After naming as A or B, the subfigure is named as a and b, which looks odd. My suggestion is to separate all figures and then just name them A, B, and C.

Response 1: Thank you very much for your responsible comments.

Line 149, 175 and 178: According to your suggestion, we split the original figure 1 into the current version of figure 1, figure 2, figure 3 and supplementary figure 1, and the clarity and quality of these figures have been significantly improved.

Point 2: The same problem happens in figure 2. Please separate all figures. In Figure 2B, font B looks bigger than the figure itself.

Response 2: Thank you for your constructive suggestions.

Line 198, 221 and 224: According to your suggestion, we split the original figure 2 into the current version of figure 4, figure 5 and figure 6, and the clarity and quality of these figures have been significantly improved.

Point 3: Figure 4 is very small. Gene's name looks more prominent than the graph itself. My suggestion is to make figure potraite instead of landscape ans add only 2 genes side by side. What is the control in this figure. Make all control as 1 or 0. It should be the same.

Response 3: Thank you very much for your kindly advice.

Line 283: Based on your suggestion, we selected two representative charts from the original figure 4 and combined them side by side. More details can be shown, and the optimized original figure 4 has been uploaded as supplementary figure 2. The clarity and quality of these figures have been significantly improved. All controls are 1 in qRT-PCR calculation, and these FPKM values in RNA-seq data are raw data without standardization.

Point 4: Consider dividing the discussion part into subsections.

Response 4: Thank you for your constructive suggestions.

Line 365, 381, 391, 405-406, 424, 439-440, 451, 475: The discussion part is divided into eight subsections.

Point 5 (1): L11 in the response to TO in response to.

Response 5 (1): Thank you for your careful correction.

Line 13: This point has been changed correctly.

Point 5 (2): L12 What is JAZ? Write down the full form.

Response 5 (2): Thank you for your careful inspection.

Line 14: The full name of JAZ is JASMONATE ZIM domain. The full name has been added correctly.

Point 5 (3): L17 MeJA and SA. TRV2 full form?

Response 5 (3): Thank you very much for your careful check.

Line 14, 19-21: The full names of these abbreviations have been added to the correct location, and has checked the full text.

Point 5 (4): L20 may be closely to might be closely.

Response 5 (4): Thank you for your careful correction.

Line 24: This point has been corrected correctly.

Point 5 (5): L17, L19, L22, L25, L27, L58, and L326, L635, L637 leaf spot to leaf spots.

Response 5 (5): Thank you very much for your kindly advice.

Line 3, 21, 23, 26, 30, 31, 65, 89, 96, 103, 119, 350, 446, 451, 456, 458, 459, 464, 631, 668, 679 and 681: We have made correct changes and examined them carefully in the full text. Meanwhile, according to the suggestion of reviewer 1, we replaced all 'gray leaf spot' in the manuscript with 'grey leaf spots'.

Point 5 (6): L33 continually at-tacked to continually attacked.

Response 5 (6): Thank you very much for your responsible comments.

Line 38: This point has been corrected correctly.

Point 5 (7): At L35 it is written anti-bacterial and at L347 its is antibacterial. Please be consistant.

Response 5 (7): Thank you for your careful inspection.

Line 40: This point has been corrected correctly.

Point 5 (8): L36 hormones and so on are all? This line is strange.

Response 5 (8): Thank you very much for your suggestions.

Line 39-41: In order to make the reader understand more clearly, the sentence has been rewritten as 'There are many ways for plants to protect themselves, such as programmed cell death, secretion of antibacterial substances, and production of endogenous hormones (salicylic acid, jasmonic acid, ethylene)'.

Point 5 (9): L39 and connect with each other to form a network to and connect to form a network.

Response 5 (9): Thank you very much for your responsible comments.

Line 44: This point has been corrected correctly.

Point 5 (10): L40 further eluci-dating to further elucidating.

Response 5 (10): Thank you for your careful correction.

Line 46: This point has been corrected correctly.

Point 5 (11): L42 ZIM full form?

Response 5 (11): Thank you very much for your careful check.

Line 48: The full name of ZIM is Zinc-finger protein expressed in Inflorescence Meristem. The full name has been added correctly.

Point 5 (12): L62 TIFY full form? At L42 ZIM domain is JAZ however, at L62 ZIM domain is TIFY! Which is true?

Response 5 (12): Thank you for your careful inspection.

Line 48: The full name of TIFY is containing a highly conserved TIFY motif.

Line 69, 70 and 73: We have changed ‘ZIM ’ to ‘TIFY’ and has checked the full text

Point 5 (13): L71 this may due to this may be due.

Response 5 (13): Thank you very much for your responsible comments.

Line 79: This point has been corrected correctly.

Point 5 (14): L78 may have a negatively regulate to may have negatively regulated.

Response 5 (14): Thank you for your constructive suggestions.

Line 86-87: According to your suggestion and the comments of reviewer 1, we have changed this sentence to 'these results suggest that the expression of JAZ protein in plants may reduce plant defense against diseases'.

Point 5 (15): L80 most serious plant diseases to most severe plant diseases.

Response 5 (15): Thank you for your careful correction.

Line 89: This point has been corrected correctly.

Point 5 (16): L105 significantly increases what?

Response 5 (16): Thank you very much for your responsible comments.

Line 114: We changed this sentence to 'which significantly increased the expression of these genes'.

Point 5 (17): L114-115 Full form and weblink of HMMER, SMART, and CDD, also write down the date when you have accessed them latest.

Response 5 (17): Thank you very much for your suggestions.

Line 123-127: We have added the full names of these abbreviations in place and wrote down the date on which they were last accessed.

Point 5 (18): L124 Write down molecular weight instead of MW first time.

Response 5 (18): Thank you for your careful correction.

Line 138: This point has been corrected correctly.

Point 5 (19): L134 in Section 2.2 for phylogenetic tree author must have used amino acid instead of nucleotides so write down all gene name nomal not italic as they are candidate proteins not nucleotide.

Response 5 (19): Thank you very much for your responsible comments.

Line 153-177: This point has been corrected correctly.

Point 5 (20): L209 What is FPKM and TFGD full form?

Response 5 (20): Thank you very much for your responsible comments.

Line 229-230: We have added the full names of these abbreviations in place.

Point 5 (21): L245 On the basis of to Based on.

Response 5 (21): Thank you very much for your careful check.

Line 266: This point has been corrected correctly.

Point 5 (22): L259 the expression of SlJAZ genes may be regulated by JA and SA to JA and SA may regulate the expression of SlJAZ genes.

Response 5 (22): Thank you very much for your responsible comments.

Line 282: This point has been corrected correctly.

Point 5 (23): L301 boxes indicate to boxes indicates.

Response 5 (23): Thank you for your careful inspection.

Line 324: This point has been corrected correctly.

Point 5 (24): L440 Pattern of the immune to Hypothetical pattern of the immune.

Response 5 (24): Thank you for your careful correction.

Line 474: This point has been corrected correctly.

Point 5 (25): L444 Full form of DELLA.

Response 5 (25): Thank you very much for your responsible comments.

Line 479: The full form of DELLA is highly conserved N-terminal DELLA motif.

Point 5 (26): Write down full form of software or webtool you have used in material method secton along with the date when you have used those.

Response 5 (26): Thank you very much for your suggestions.

Line 137-138, 155-156, 511, 513, 524, 529-531, 537-539, 541, 548-449, 552, 558, 564, 580-582, 630, 635, and 660: We have made a comprehensive inspection of the problem in the full text and made a serious revision.

Point 5 (27): L631 SlJAZ genes are vary widely to SlJAZ genes vary widely.

Response 5 (27): Thank you very much for your kindly advice.

Line 474: This point has been corrected correctly.

Point 5 (28): L636 that other regulatory genes to that other regulatory gene or to those other regulatory genes.

Response 5 (28): Thank you very much for your responsible comments.

Line 680: Thank you for your careful correction.

Thank you again for your comments and hope to learn more from you!

Round 2

Reviewer 3 Report

I am more than happy with the author's responses. The manuscript looks much better and refined now and hence can be accepted in its current form.